# Determinants of preterm survival in a tertiary hospital in Ghana: A ten-year review

Evans Kofi Agbeno[1]*, Joseph Osarfo[2], Joyce Ashong[3], Betty Anane-Fenin[1], Emmanuel Okai[3], Anthony Amanfo Ofori[1], Mohammed Aliyu[4], Douglas Aninng Opoku[4], Sebastian Ken-Amoah[1], Joycelyn A. Ashong[1], Hora Soltani[5]

1 Department of Obstetrics and Gynaecology, School of Medical Sciences, University of Cape Coast, Cape Coast, Ghana/Cape Coast Teaching Hospital, Cape Coast, Ghana, 2 Ghana Health Service, Asante Mampong Municipal hospital, Ashanti Region, Ghana, 3 Department of Paediatrics, School of Medical Sciences, University of Cape Coast, Cape Coast, Ghana/Cape Coast Teaching Hospital, Cape Coast, Ghana, 4 School of Public Health, Kwame Nkrumah University of Science and Technology, Kumasi, Ghana, 5 Faculty of Health and Wellbeing, Sheffield Hallam University, Sheffield, United Kingdom

☯ These authors contributed equally to this work.

* evans.agbeno@ucc.edu.gh

**Data Availability Statement:** All relevant data are within the manuscript.

## Abstract

### Background

Prematurity (gestational age <37 completed weeks) accounts for the majority of neonatal deaths worldwide and most of these occur in the low-resource countries. Understanding factors that determine the best chances of preterm survival is imperative in order to enhance the care of neonates and reduce adverse outcomes in such complicated births.

### Aim

This was to find out the proportions of preterm babies who survived at the Special Care Baby Unit (SCBU) in the Cape Coast Teaching Hospital (CCTH) and the factors which influenced their survival.

### Method

This was a retrospective review of data on all the live preterm babies seen at the SCBU of CCTH from 2010 to 2019. Data on 2,254 babies that met the inclusion criteria were extracted. Descriptive statistics were generated and tests of association done with chi-square and multivariable logistic regression.

### Outcome

The main outcome measure was the proportion of live preterm neonates who were discharged after SCBU admissions.

### Results

The CCTH had a total of 27,320 deliveries from 2010 to 2019. Of these, 1,282 were live preterm births, giving a prevalence of live preterm babies over the ten-year period of 4.7% (1,282/27,320). An increasing trend in prevalence was observed with 2019 recording the

**Funding:** The authors received no specific funding for this work.

**Competing interests:** The authors have declared that no competing interests exist.

highest at 9% (271/3027). Most (48.8%) of the deliveries were vaginal, 39.2% were by caesarean section (CS); the mode of birth for 12% of the women were not documented. The mean gestational age was 31.8 (±2.77) weeks. Of the birth weights documented, <1000g babies accounted for 11.9%, 1000–1499g babies made up 34.8%, while 1500g to 2499g babies accounted for 42.6%. The babies with weights >2500g made up only 3.7%. The average length of hospital stay was 8.3 (±9.88) days. Regarding the main outcome variable, 67.6% were discharged alive, 27.6% died and 4.9% were unaccounted for due to incomplete documentation. Factors which influenced survival were: birth weight (p <0.001); gestational age (p <0.001); mode and place of delivery (p <0.001 for both); APGAR scores at 1st and 5th minutes (p <0.001); and length of stay at the SCBU (p <0.001). No association was found for sex of the baby, maternal age and parity.

## Conclusion

This study shows the possibility of achieving good preterm survival rates through the provision of specialised neonatal care, even in resource-constrained countries. This provides an updated benchmark for clinical decision-making and antenatal counselling. It also highlights the problem of inadequate data capture in our part of the world, which needs considerable improvement.

## Introduction

Preterm births (births before 37 weeks) have gained public health prominence because of the unique short- and long-term complications that affect neonatal survival and quality of life [1, 2]. An estimated 15 million preterm births occur annually globally, translating into 1 in 10 deliveries, with about 81% occurring in South Asia and Sub-Saharan Africa [3–6]. Prematurity is the major cause of low-birth weight [6], and is also the first of the six leading causes of under-5 mortality, accounting for about a million deaths per year globally [5]. Preterm births underlie many complications including respiratory distress syndrome, bronchopulmonary dysplasia, intraventricular haemorrhage, periventricular leukomalacia, anaemia, infections (sepsis), neurodevelopmental disabilities and chronic conditions such as asthma and diabetes mellitus later in life. They also place physical, psychological, and financial stress to the affected families [4, 7].

Increased use of assisted reproductive technology resulting in multiple pregnancies and pregnancy in advanced maternal age leading to complicated pregnancies are the main causes of preterm delivery in high-income countries. In contrast, preterm births in low-income countries are significantly associated with infections, malaria, HIV, and adolescent pregnancy, among others [3]. Many effective interventions including antenatal corticosteroids use, maternal antenatal magnesium sulphate administration, antibiotics for preterm prelabour rupture of membranes (PPROM) and the use of surfactants have led to improved preterm survival [8], but this has been disproportionately higher in the high-income countries compared to those in low and middle income countries (LMIC).

Aside organ maturity, which is itself a function of gestational age and birthweight, preterm survival is aided by effective health systems support. It has been shown that establishing specialized neonatal care units in district hospitals, provision of early respiratory support, adequate antenatal care, use of first trimester biomarkers and low-dose aspirin to reduce the risk

of pre-eclampsia, administration of progesterone and procedures such as cerclage for cervical insufficiency significantly prolong pregnancy and increase chances of survival [9–16].

In Ghana, 128,000 babies are born preterm annually and 8,400 of these die before age five due to direct complications of prematurity [14]. The doubled rate of preterm birth in Ghana's highest referral centre (Korle Bu Teaching Hospital) from 9.3% to 18.9% over the past fifteen years (2003 to 2019) is a public health concern [15, 16], and associations between preterm births and advanced maternal age (>35 years), suboptimal antenatal care, preeclampsia, antepartum haemorrhage, and PPROM have been reported [17, 18]. To authors knowledge, no study has been conducted to assess survival rates of preterm babies admitted for specialized care and the factors associated with their survival. Previous studies in Ghana sought to establish hospital-based incidence of preterm births and its association with some maternal characteristics and obstetric complications [15, 16].

The present study, a retrospective review of existing data on preterm infants admitted to the specialized care baby unit (SCBU) of the Cape Coast Teaching Hospital (CCTH), is the first to report on survival rates of preterm babies at such a unit in Ghana. Knowledge of factors affecting survival of preterm neonates in a country with health system challenges, such as Ghana, can guide clinicians better on the timing of delivery in mild-to-moderately complicated pregnancies. This will enable a good balance between fetal/neonatal and maternal outcomes when compromising decisions have to be made [6].

## Materials and methods

### Study type

This was a ten-year review of available data on all documented preterm babies admitted into the SCBU OF CCTH from 1st January, 2010 to 31st December, 2019.

### Study site

The CCTH is the only tertiary health facility offering specialised services to the inhabitants of the Central, Western, parts of the Ashanti and Greater Accra regions of Ghana. It has a specialised neonatal care unit which lacks some of the logistics of a neonatal intensive care unit (NICU) such as mechanical ventilators, adequate and fully-functioning non-invasive ventilatory support, perfusers, and blood gas analysers. Hence, the unit is more appropriately termed a 'Special Care Baby Unit' (SCBU). It serves as the main point of call for all preterm babies delivered within the hospital and its catchment areas. As such, it is mandatory for all preterm babies born alive at CCTH to be sent to the SCBU for assessment and further management. The unit has six paediatricians who are complimented by other health cadres.

### Study population

All preterm babies managed at the SCBU from 1st January, 2010 to 31st December, 2019.

### Inclusion criteria

All live preterm babies who were admitted to the SCBU of CCTH within the study period with adequate available data (described as a minimum of 85% of required data).

### Exclusion criteria

Preterm babies pronounced clinically dead at the time of arrival at the SCBU and babies without adequate documentation. Also excluded were preterm babies who did not pass through SCBU.

## Data collection/handling

Data collection was from August 2018 to December 2019 and was conducted by two special-ists- in-training (paediatrics and obstetrics/gynaecology). The principal investigator (EKA) trained the data collectors over three days on key components of the study including the objectives, inclusion/exclusion criteria and the study variables to be captured using a data extraction template.

Data on maternal and neonatal characteristics including maternal age, parity, mode and year of delivery, sex and weight of the baby, gestational age, 1st and 5th minute APGAR scores, length of stay at the SCBU and outcome (whether discharged alive or demised) were collected from primary sources such as the admissions and discharges register of the SCBU, theatre and recovery unit registers and the birth register at the maternity ward. The data were anonymized by taking off personal identifiers and replacing them with study identity numbers. The data were entered into an Excel spreadsheet, cleaned to exclude incomplete entries and exported into SPSS version 22 for analysis. Data on 20% of the study participants were cross-checked with the primary sources for quality control.

Means and frequencies were computed and presented in tables and charts. Chi-square tests and logistic regression methods were employed to test for associations and the strength thereof between the dependent variable (preterm survival defined as 'being discharged home alive') and independent variables.

## Ethical clearance

Ethical clearance for the study was granted by the Ethical Review Board of Cape Coast Teaching Hospital with reference number CCTHERC/EC/2018/20. Since this was essentially a retrospective study using secondary data, it was not possible to obtain informed consent for the bulk of participants. Participants' informed consent was therefore not sought. This was made clear in the application for ethical clearance.

## Results

### Demographic and clinical characteristics of study participants

The study included a total of 2,254 preterm babies seen at the SCBU from 2010 to 2019. The CCTH had a total of 27,320 deliveries within that same period, out of which 1,282 were live preterm births. A further 623 preterms were from different referring facilities while data on the place of birth for another 349 babies were not documented. Thus, the prevalence of live preterm babies over the 10-year period for deliveries at CCTH alone, was 4.7% (1,282/27,320).

Of those with the mode of delivery documented among the 2254 babies, 55.5% (1,100/1983) were vaginal and 44.5% (883/1983) were by caesarean section (see Table 1). The ages of the babies in weeks ranged from 20 to 36 with a mean(SD) of 31.8 (± 2.8). The babies stayed at the SCBU between 30 minutes and 188 days with a mean of 8.3 (SD ± 9.9) days. There were similar proportions of both male and female babies.

The proportion of preterm babies that survived and were discharged home alive was 67.6% (1,523/2254) while 27.6% (621/2254) died. There was no survival outcome data for 110 of the babies. An increasing trend in prevalence of preterm infants is observed, with 2019 recording the highest of 9% (271/3027) [see Fig 1]. The highest survival rate was recorded in 2012 (94.1%). [see Fig 2].

The age of the mothers ranged from 14 to 47 and a mean of 27.5 (SD ± 7.0) with the majority (50.5%, 837/1658) in the 20–30-year age group [see Table 1].

**Table 1. Demographic and clinical characteristics of study participants.**

| Variable | n (%) |
|---|---|
| **Sex of baby (N = 2,234)** | |
| male | 1,115 (49.9) |
| female | 1,119 (50.1) |
| **Maternal Age (yrs) (N = 1,658)** | |
| Mean maternal age (SD) | 27.5 (7.0) |
| < 20 | 259 (15.6) |
| 20–30 | 837 (50.5) |
| 31–40 | 520 (31.4) |
| > 40 | 42 (2.5) |
| **Parity (N = 1580)** | |
| 1 | 661 (41.8) |
| 2–4 | 660 (41.8) |
| ≥ 5 | 259 (16.4) |
| **Weight of Baby (g) (N = 2098)** | |
| Extremely Low (< 1000g) | 268 (12.8) |
| Very Low (1000–1499g) | 785 (37.4) |
| Low Birth Weight (1500–2499g) | 961 (45.8) |
| Normal Weight (≥ 2500g) | 84 (4.0) |
| **APGAR Score at 1 minute (N = 1085)** | |
| 1 | 121 (11.2) |
| 2 | 130 (12.0) |
| 3 | 104 (9.6) |
| 4 | 158 (14.6) |
| 5 | 168 (15.5) |
| 6 | 203 (18.7) |
| 7 | 160 (14.7) |
| 8 | 41 (3.8) |
| **APGAR Score at 5 minutes (N = 1083)** | |
| 1 | 22 (2.0) |
| 2 | 68 (6.3) |
| 3 | 81 (7.5) |
| 4 | 117 (10.8) |
| 5 | 160 (14.8) |
| 6 | 175 (16.2) |
| 7 | 225 (20.8) |
| 8 | 187 (17.3) |
| 9 | 47 (4.2) |
| 10 | 1 (0.1) |
| **Gestational Age of Baby (weeks) (N = 1319)** | |
| Mean gestational age (SD) | 31.8 (2.8) |
| < 28 | 83 (6.3) |
| 28 | 113 (8.6) |
| 29 | 75 (5.7) |
| 30 | 137 (10.4) |
| 31 | 121 (9.2) |
| 32 | 223 (16.9) |
| 33 | 169 (12.8) |

(*Continued*)

**Table 1.** (Continued)

| Variable | n (%) |
| --- | --- |
| 34 | 172 (13.0) |
| 35 | 141 (10.7) |
| 36 | 85 (6.4) |
| **Length of Stay at SCBU (days) (N = 2067)** | |
| Mean (SD) | 8.3 (9.9) |
| 0–5 | 928 (44.9) |
| 6–10 | 592 (28.6) |
| 11–15 | 294 (14.3) |
| 16–20 | 102 (4.9) |
| 21–25 | 60 (2.9) |
| 26–30 | 37 (1.8) |
| 31–35 | 23 (1.1) |
| > 35 | 31 (1.5) |
| **Year of Delivery (N = 2254)** | |
| 2010–2014 | 670 (29.7) |
| 2015–2019 | 1,584 (70.3) |
| **Survival of Preterm Neonates (N = 2144)** | |
| Death | 621 (29.0) |
| Discharged/Alive | 1,523 (71.0) |
| **Mode of Delivery (N = 1983)** | |
| CS | 883 (44.5) |
| SVD | 1,100 (55.5) |
| **Referral Point/Place of delivery (N = 1905)** | |
| CCTH | 1,282 (67.3) |
| Others | 623 (32.7) |

## Factors influencing survival of preterm babies

Table 2 shows the factors influencing the survival of preterm babies.

Survival of the preterm neonate was independent of the sex of the baby, maternal age and parity. However, the weight of a baby (p <0.001), referral point (p <0.001), mode of delivery (p <0.001), APGAR score at 1 minute (p <0.001), APGAR score at 5 minutes (p<0.001), length of stay at SCBU (p<0.001) and gestational age (p<0.001) were significantly associated with survival of preterm babies by Chi square analysis.

Table 3 shows the results of the logistic regression analysis of factors associated with the survival outcomes of preterm babies. In the multivariate analysis, each variable considered in the univariate analysis was adjusted for using the other independent variables.

Preterm infants from spontaneous vaginal delivery had lower odds of survival compared to those delivered by caesarean section (OR 0.57, 95% CI: 0.46, 0,70; p<0.001). This difference was, however, obliterated in the multivariate analysis adjusting for birthweight, referral point, gestational age, length of stay and Apgar scores.

While gestational ages ≥28 weeks showed increasingly higher odds of survival in the univariate analysis, only gestational ages ≥32 weeks were shown to aid preterm survival in the multivariate analysis. Increasing baby weight was also demonstrated to be relevant for preterm survival.

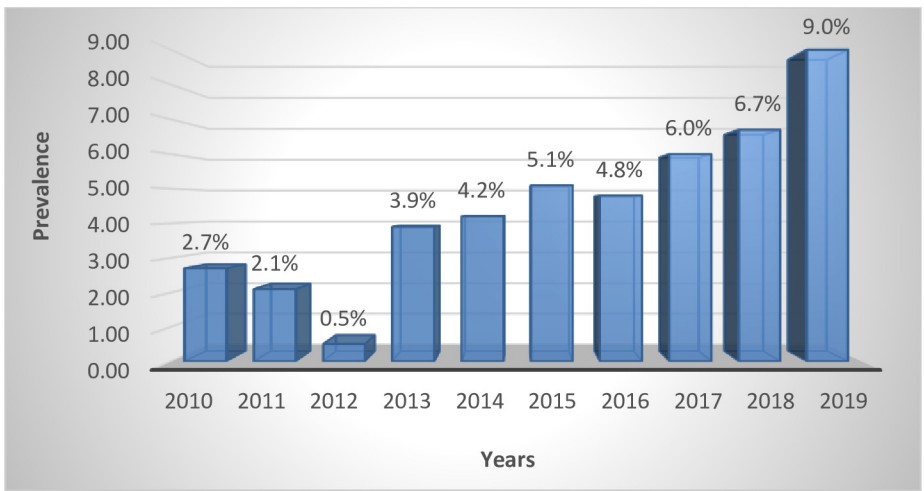

**Fig 1. Prevalence of preterm babies for the 10 years period at CCTH.**

## Discussion

Knowledge of current neonatal mortality rates is fundamental for health professionals in the care and management of high-risk obstetric patients. Obstetricians use this information in the decision-making process when confronted with problems such as preterm labour and medical complications that may necessitate early delivery or transfer to a perinatal centre. Expectant management decisions are based on the provider's perceptions of neonatal survival. Information of current preterm mortality rates is also helpful when counselling families regarding potential outcomes associated with early delivery.

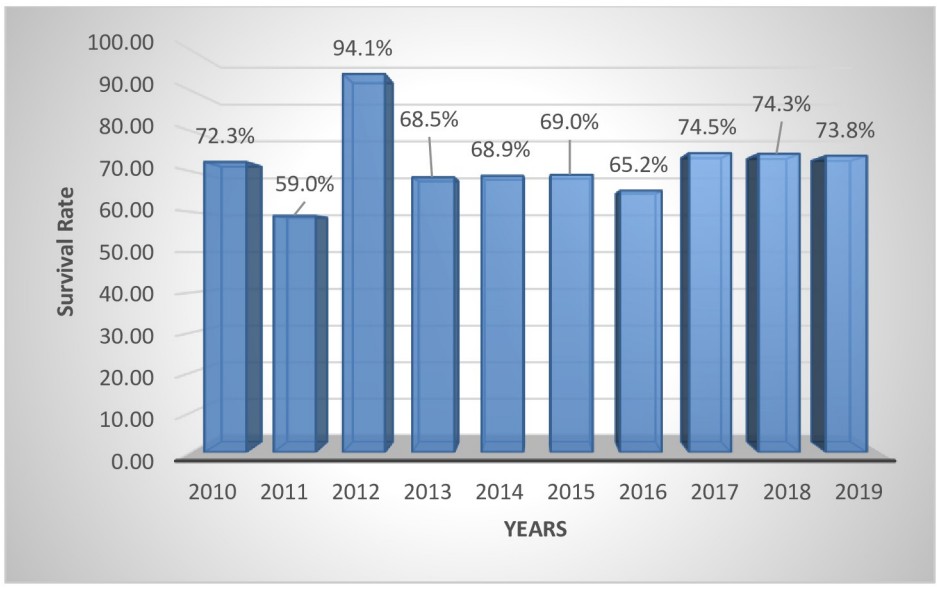

**Fig 2. Survival rate of preterm babies for the 10 years.**

**Table 2. Factors influencing the survival outcomes of preterm babies.**

| Factors | Survival | | *P-Value |
|---|---|---|---|
| | Discharged alive n (%) | Died n (%) | |
| **Sex of Baby** | | | 0.321 |
| Male | 746 (70.3) | 315 (29.7) | |
| Female | 771 (72.3) | 296 (27.7) | |
| **Age of Mother (years)** | | | 0.598 |
| < 20 | 184 (74.2) | 64 (25.8) | |
| 20–30 | 587 (73.7) | 210 (26.3) | |
| 31–40 | 355 (70.7) | 147 (29.3) | |
| > 40 | 32 (76.2) | 10 (23.8) | |
| **Parity** | | | 0.717 |
| 0–1 | 456 (72.4) | 174 (27.6) | |
| 2–4 | 470 (74.2) | 163 (25.8) | |
| ≥ 5 | 182 (74.3) | 63 (25.7) | |
| **Mode of Delivery** | | | <0.001 |
| CS | 656 (77.4) | 191 (22.6) | |
| SVD | 698 (66.2) | 357 (33.8) | |
| **Referral Point** | | | <0.001 |
| CCTH | 932 (76.1) | 292 (23.9) | |
| Others | 379 (62.9) | 224 (37.1) | |
| **Weight of Baby** | | | <0.001 |
| Extremely Low (< 1000g) | 69 (26.4) | 192 (73.6) | |
| Very Low (1000–1499g) | 512 (68.5) | 235 (31.5) | |
| Low Birth Weight (1500–2499g) | 786 (86.3) | 125 (13.7) | |
| Normal Weight (≥ 2500g) | 68 (86.1) | 11 (13.9) | |
| **APGAR Score at 1 minute** | | | <0.001 |
| 1 | 61 (51.3) | 58 (48.7) | |
| 2 | 73 (58.4) | 52 (41.6) | |
| 3 | 71 (73.2) | 26 (26.8) | |
| 4 | 116 (76.3) | 36 (23.7) | |
| 5 | 138 (87.3) | 20 (12.7) | |
| 6 | 163 (83.6) | 32 (16.4) | |
| 7 | 141 (90.4) | 15 (9.6) | |
| 8 | 36 (94.7) | 2 (5.3) | |
| **APGAR Score at 5 minutes** | | | <0.001 |
| 1 | 6 (28.6) | 15 (71.4) | |
| 2 | 31 (45.6) | 37 (54.4) | |
| 3 | 45 (56.2) | 35 (43.8) | |
| 4 | 71 (64.0) | 40 (36.0) | |
| 5 | 117 (76.5) | 36 (23.5) | |
| 6 | 137 (82.8) | 28 (17.2) | |
| 7 | 185 (86.0) | 30 (14.0) | |
| 8 | 164 (89.7) | 19 (10.3) | |
| 9 | 41 (95.3) | 2 (4.7) | |
| **Length of Stay at NICU** | | | <0.001 |
| 0–5 | | | |
| 6–10 | 473 (51.0) | 455 (49.0) | |
| 11–15 | 502 (84.8) | 90 (15.2) | |

(*Continued*)

**Table 2.** (Continued)

| Factors | Survival | | *P-Value |
|---|---|---|---|
| | **Discharged alive n (%)** | **Died n (%)** | |
| 16–20 | 262 (89.4) | 31 (10.6) | |
| 21–25 | 90 (88.2) | 12 (11.8) | |
| 26–30 | 55 (91.7) | 5 (8.3) | |
| 31–35 | 32 (86.5) | 5 (13.5) | |
| > 35 | 20 (87.0) | 3 (13.0) | |
| **Gestational Age of Baby (Weeks)** | | | <0.001 |
| < 28 | 18 (22.0) | 64 (78.0) | |
| 28 | 52 (47.3) | 58 (52.7) | |
| 29 | 42 (57.5) | 31 (42.5) | |
| 30 | 82 (62.1) | 50 (37.9) | |
| 31 | 88 (75.9) | 28 (24.1) | |
| 32 | 163 (75.8) | 52 (24.2) | |
| 33 | 137 (84.6) | 25 (15.4) | |
| 34 | 153 (91.6) | 14 (8.4) | |
| 35 | 121 (91.7) | 11 (8.3) | |
| 36 | 77 (91.7) | 7 (8.3) | |
| **Year of Delivery** | | | 0.212 |
| 2010–2014 | 425 (69.1) | 190 (30.9) | |
| 2015–2019 | 1098 (71.8) | 431 (28.2) | |

*Analysis by Chi Square.

This study reports on the mortality/survival rates in the neonatal unit in Cape Coast over a 10-year period (2010–2019) and some associating factors. As the only tertiary neonatal facility, west of the capital Accra, the data provides an important source of local epidemiological information which has previously not been described. The derived local centre-specific information about preterm infants would be invaluable to support clinical decision-making and inform budgetary allocation, as well as facilitate antenatal counselling in high-risk infants.

The present study showed that the survival rate on discharge of preterm infants was highest in the higher gestational age (GA) ranges, ranging from 91.6% for 34weeks, 84% at 33weeks, 76% at 32weeks, 75% at 31 weeks, 62% at 30weeks, 56% at 29 weeks, 47% at 28weeks and significantly dropping to 28% at GA less than 28weeks. The overall survival rate of the studied cohort was 67.4%; a percentage which is much more encouraging than that quoted for low-income countries in the 'Born Too Soon Preterm Birth Group' review [19], but significantly lower than that described in multicentre studies from high-income countries, such as the MOSAIC study, which documented 89.5% of survival at gestations less than 32weeks [19–21], and mortality rates of 1.4% at gestations less than 37 weeks in the United States [22]. The 'Born Too Soon Preterm Birth Action Group' reported about 50% survival for neonates born around a gestational age of 34 weeks, and 30% among preterms born at less than 32 weeks [19, 21]. Malawi has described similar survival rates of 68% and 19% [23], while an overall singleton preterm survival rate of 38.4% has been reported in Nigeria [24].

We speculate that the higher survival rate that was achieved at the SCBU, compared to other units in similar resource-constrained countries, was due to the conscious implementation of certain interventions and policies, both locally and nationally. These include the increasing availability of trained and skilled personnel across both the Obstetrics and

**Table 3. Univariate and multivariate logistic regression analysis of factors associated with survival outcomes of preterm babies.**

| Factors | Univariate | | *Multivariate | |
|---|---|---|---|---|
| | Unadjusted OR (95%CI) | P-value | Adjusted OR (95% CI) | P-value |
| **Mode of Delivery** | | | | |
| CS *Ref_* | 1.00 | | 1.00 | |
| SVD | 0.57 (0.46–0.70) | <0.001 | 1.21 (0.77–1.89) | 0.414 |
| **Referral Point** | | | | |
| CCTH *Ref_* | 1.00 | | 1.00 | |
| Others | 0.53 (0.43–0.65) | <0.001 | 0.35 (0.18–0.70) | 0.003 |
| **Weight of Baby** | | | | |
| Extremely Low (< 1000g) *Ref_* | 1.00 | | 1.00 | |
| Very Low (1000–1499g) | 6.06 (4.42–8.31) | <0.001 | 2.90 (1.27–6.62) | 0.011 |
| Low Birth Weight (1500–2499g) | 17.50 (12.53–24.43) | <0.001 | 7.58 (3.08–18.68) | <0.001 |
| Normal Weight (≥ 2500g) | 17.20 (8.59–34.43) | <0.001 | 7.08 (1.65–30.37) | 0.008 |
| **APGAR Score at 1 minute** | | | | |
| 1 *Ref_* | 1.00 | | 1.00 | |
| 2 | 1.33 (0.81–2.21) | 0.263 | 0.56 (0.17–1.86) | 0.347 |
| 3 | 2.60 (1.46–4.62) | 0.001 | 0.84 (0.19–3.71) | 0.818 |
| 4 | 3.06 (1.82–5.15) | <0.001 | 0.23 (0.04–1.22) | 0.085 |
| 5 | 6.56 (3.63–11.85) | <0.001 | 0.33 (0.05–2.09) | 0.241 |
| 6 | 4.84 (2.87–8.17) | <0.001 | 0.13 (0.02–1.01) | 0.051 |
| 7 | 8.94 (4.70–16.99) | <0.001 | 0.45 (0.04–5.13) | 0.521 |
| 8 | 17.11 (3.94–74.32) | <0.001 | 0.15 (0.00–108.01) | 0.576 |
| **APGAR Score at 5 minutes** | | | | |
| 1 *Ref_* | 1.00 | | 1.00 | |
| 2 | 2.09 (0.73–6.05) | 0.172 | 2.37 (0.59–9.45) | 0.221 |
| 3 | 3.21 (1.13–9.14) | 0.028 | 5.55 (1.14–27.09) | 0.034 |
| 4 | 4.44 (1.60–12.34) | 0.004 | 6.78 (1.21–38.11) | 0.030 |
| 5 | 8.13 (2.94–22.48) | <0.001 | 31.06 (4.39–219.74) | 0.001 |
| 6 | 12.23 (4.37–34.27) | <0.001 | 46.98 (5.63–391.92) | <0.001 |
| 7 | 15.42 (5.55–42.85) | <0.001 | 85.17 (8.12–893.75) | <0.001 |
| 8 | 21.58 (7.48–62.24) | <0.001 | 34.39 (2.52–468.50) | 0.008 |
| 9 | 51.25 (9.31–282.26) | <0.001 | 158.99 (0.20–123644.30) | 0.136 |
| **Length of Stay at NICU** | | | | |
| 0–5 *Ref_* | 1.00 | | 1.00 | |
| 6–10 | 5.37 (4.14–6.95) | <0.001 | 8.56 (4.85–15.12) | <0.001 |
| 11–15 | 8.13 (5.48–12.05) | <0.001 | 16.90 (8.07–35.37) | <0.001 |
| 16–20 | 7.21 (3.90–13.36) | <0.001 | 33.28 (8.96–123.54) | <0.001 |
| 21–25 | 10.58 (4.20–26.67) | <0.001 | 113.62 (12.91–1000.28) | <0.001 |
| 26–30 | 6.16 (2.38–15.94) | <0.001 | 36.87 (5.81–233.99) | <0.001 |
| 31–35 | 6.41 (1.89–21.73) | 0.003 | 21.89 (3.05–156.93) | 0.002 |
| > 35 | 5.00 (1.90–13.14) | 0.001 | 88.14 (8.94–869.29) | <0.001 |
| **Gestational Age of Baby (Weeks)** | | | | |
| < 28 *Ref_* | 1.00 | | 1.00 | |
| 28 | 3.19 (1.68–6.06) | <0.001 | 1.57 (0.47–5.23) | 0.460 |
| 29 | 4.82 (2.39–9.69) | <0.001 | 1.35 (0.37–4.91) | 0.646 |
| 30 | 5.83 (3.11–10.95) | <0.001 | 2.12 (0.67–6.70) | 0.200 |
| 31 | 11.17 (5.70–21.93) | <0.001 | 3.30 (0.99–10.95) | 0.052 |
| 32 | 11.15 (6.06–20.49) | <0.001 | 4.23 (1.34–13.36) | 0.014 |

*(Continued)*

**Table 3.** (Continued)

| Factors | Univariate | | *Multivariate | |
|---|---|---|---|---|
| | Unadjusted OR (95%CI) | P-value | Adjusted OR (95% CI) | P-value |
| 33 | 19.48 (9.92–38.25) | <0.001 | 6.71 (1.96–22.97) | 0.002 |
| 34 | 38.86 (18.23–82.83) | <0.001 | 12.95 (3.59–46.67) | <0.001 |
| 35 | 39.11 (17.42–87.83) | <0.001 | 11.28 (3.05–41.74) | <0.001 |
| 36 | 39.11 (15.37–99.51) | <0.001 | 21.79 (5.04–94.20) | <0.001 |

Ref = Reference category.

*Each variable was adjusted for using the other independent variables presented.

Gynaecology, and Paediatrics departments, use of maternal antenatal steroids and magnesium sulphate for fetal lung maturity and neuroprotection respectively; paediatricians attending high-risk deliveries, and continued improvement in cost-effective care, such as the provision of warmth, breastfeeding support, basic care for infections, and the increasing but challenging use of non -invasive respiratory support for preterm babies with breathing difficulties.

When considering survival by gestational age, it is observed that the differences become even more stark. The survival rate in the MOSAIC study which assessed mortality in ten European countries in the year 2003 and included premature infants between 24 and 31weeks reported survival rates of 58.2% for GAs of 24 to 27 weeks, and 92.4% for 28 to 31 weeks. Survival of preterm babies is comparatively very high in resource rich countries due to the readily-available technological equipment. These logistics, among other factors, have contributed to making the survival at 34 weeks equivalent to that of a full term baby in the United Kingdom [25].

Multiple factors are known to be associated with neonatal mortality and morbidity. In our current study, it was identified that low birth weight and low Apgar score at 5 minutes have a huge impact. Gestational age and weight at birth are important determinants of preterm survival, with mortalities being inversely proportional to gestational age [20, 26]. With each additional week of gestation and 100-gram increase in birth weight, the mortality risk in neonates reduces by an appreciable margin [26]. This remains true with our study, which also shows an unacceptably high overall mortality rate of 27.7%, contributed largely by deaths of babies who are less than 28 weeks (79.3%). Mortality rates are 52% at 28 weeks, 43.2% at 29weeks, 38.7% at 30weeks, reducing to 8.1% at 36weeks. We believe that delays in reaching the SCBU, lack of standardized antenatal care, unavailability of equipment, and appropriate laboratory support, contribute significantly to the high mortality in these preterm neonates.

Regarding neonatal mortalities by birth weight, our study reports increased mortality rates of 73.8%, 31.2%, and 13.9% at weights less than 1000g, at 1000–1499g, and >1500g respectively, in support of known literature. Similar to other studies, the majority of deaths reported in our study occurred in the first few days of life, particularly among the most premature babies [27, 28].

Knowledge of gestational age and birthweight-specific mortality and morbidity in individual neonatal units, particularly in tertiary centres like ours in resource-constrained countries, is very important for the strategic planning of care, personnel training and budgetary allocation as well as from the patient-satisfaction point of view. We found an increasing prevalence of live preterm birth at CCTH, which is similar to the global situation, but the fact that babies born outside the teaching hospital had 3 times lower chances of survival shows that even national budgetary support may be biased towards the tertiary centres. There must be a

conscious effort to provide basic neonatal care at the community level to improve survival in the preterm population, as highlighted by Ndelema et al [9]. With the increasing survival at limits of viability, especially in developed countries, data such as ours and others from other tertiary centres can guide decisions into resource allocation nationwide and continue to garner global support for the developing world to catch up with the more developed nations in keeping every baby born alive, no matter the gestational age, birthweight and place of birth.

Regarding the characteristics of the population, there was no significant association between sex of baby and mortality. This finding deviatesfrom existing literature that shows males are more at risk of death than females [29]. It is possible the sample size lacked sufficient power to detect statistically significant differences for this particular outcome. The high incidence of caesarean delivery can be explained by the characteristic of the hospital, which acts as a referral centre for high-risk pregnancies.

## Strengths

Due to the decade-long accumulation of data, a high number of births and a relatively large and representative sample size of preterm babies were accrued. This study gives the first description of short-term outcome of very preterm infants born in Ghana.

## Limitations

Data on gestational age was not documented for some of the babies in the Admissions and Discharges book at SCBU and this is acknowledged as a limitation. Thus, we may have erroneously and inadvertently excluded some preterm babies from this study. Information gaps in developing countries have been highlighted in a recent review on preterm births, as one of the challenges in assessing accurate epidemiological data [21]. As a result of similar underlying information gaps, it is possible the prevalence of live preterm births may have been underreported as there was no recourse to the 623 infants referred from other facilities and the 349 infants without data on place of delivery. Furthermore, the study was unable to speak to the obstetric and medical conditions that gave rise to the occurrence of the preterm births due to documentation gaps. To optimise data collection, most of our data collection tools have undergone revision this year, to incorporate clear case definitions and capture as much relevant information as possible.

Also, we could only report on hospital (SCBU) discharge outcomes. We are not privy to their outcomes after being discharged from the SCBU. In addition, lack of information on stillbirths may overestimate the survival probabilities in our cohort compared with other reported studies.

## Conclusion

In conclusion, our study characterised the survival of preterm infants from a geographically-defined area. Our data shows that preterm infants who survive the first six postnatal days have considerable chance of survival until discharge. It also emphasizes the possibility of achieving good preterm survival rates through the provision of specialised neonatal care, even in resource-constrained countries. This provides an updated benchmark for clinical decision-making and antenatal counselling. It also highlighted the problem of inadequate data capture which needs massive improvement. Further research is needed to investigate the survival of preterm babies through a well-designed prospective cohort study for both short and long-term outcomes beyond discharge from special care units. In addition, design of appropriate data collection tools are warranted to scope a more comprehensive set of contributing factors to preterm survival.

## Author Contributions

**Conceptualization:** Evans Kofi Agbeno.

**Data curation:** Joyce Ashong.

**Formal analysis:** Mohammed Aliyu, Douglas Aninng Opoku.

**Funding acquisition:** Joseph Osarfo, Joyce Ashong, Betty Anane-Fenin, Emmanuel Okai, Anthony Amanfo Ofori, Mohammed Aliyu, Douglas Aninng Opoku, Sebastian Ken-Amoah, Hora Soltani.

**Investigation:** Evans Kofi Agbeno, Joyce Ashong, Joycelyn A. Ashong.

**Methodology:** Evans Kofi Agbeno, Joseph Osarfo, Joyce Ashong, Betty Anane-Fenin, Emmanuel Okai, Anthony Amanfo Ofori, Mohammed Aliyu, Douglas Aninng Opoku, Sebastian Ken-Amoah, Joycelyn A. Ashong, Hora Soltani.

**Project administration:** Evans Kofi Agbeno, Joyce Ashong, Betty Anane-Fenin.

**Resources:** Evans Kofi Agbeno, Joseph Osarfo, Joyce Ashong, Betty Anane-Fenin, Emmanuel Okai, Anthony Amanfo Ofori, Mohammed Aliyu, Douglas Aninng Opoku, Sebastian Ken-Amoah, Joycelyn A. Ashong, Hora Soltani.

**Software:** Mohammed Aliyu, Douglas Aninng Opoku.

**Supervision:** Evans Kofi Agbeno.

**Validation:** Betty Anane-Fenin.

**Visualization:** Evans Kofi Agbeno, Joseph Osarfo, Joyce Ashong, Betty Anane-Fenin, Emmanuel Okai, Anthony Amanfo Ofori, Mohammed Aliyu, Douglas Aninng Opoku, Sebastian Ken-Amoah, Joycelyn A. Ashong, Hora Soltani.

**Writing – original draft:** Evans Kofi Agbeno, Betty Anane-Fenin, Emmanuel Okai.

**Writing – review & editing:** Evans Kofi Agbeno, Joseph Osarfo, Joyce Ashong, Betty Anane-Fenin, Emmanuel Okai, Anthony Amanfo Ofori, Mohammed Aliyu, Douglas Aninng Opoku, Sebastian Ken-Amoah, Joycelyn A. Ashong, Hora Soltani.

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
