## [Decision Letter · Decision Letter 0]

5 Nov 2020

PONE-D-20-26520

Determinants of preterm survival in a tertiary hospital in Ghana: a ten-year review

PLOS ONE

Dear Dr. Agbeno,

Thank you for submitting your manuscript to PLOS ONE. After careful consideration, we feel that it has merit but does not fully meet PLOS ONE’s publication criteria as it currently stands. Therefore, we invite you to submit a revised version of the manuscript that addresses the points raised during the review process.

We look forward to receiving your revised manuscript.

Kind regards,

Kazumichi Fujioka

Academic Editor

PLOS ONE

Journal Requirements:

2. In the ethics statement in the manuscript and in the online submission form, please provide additional information about the patient records used in your retrospective study, including: a) whether all data were fully anonymized before you accessed them; and b) the date range (month and year) during which patients' medical records were accessed. If patients provided informed written consent to have data from their medical records used in research, please include this information.

3. We note you have included a table to which you do not refer in the text of your manuscript. Please ensure that you refer to Table 2 in your text; if accepted, production will need this reference to link the reader to the Table.

Reviewers' comments:

Reviewer's Responses to Questions

**Comments to the Author**

1. Is the manuscript technically sound, and do the data support the conclusions?

Reviewer #1: Partly

Reviewer #2: Yes

2. Has the statistical analysis been performed appropriately and rigorously? 

Reviewer #1: No

Reviewer #2: Yes

3. Have the authors made all data underlying the findings in their manuscript fully available?

Reviewer #1: No

Reviewer #2: Yes

4. Is the manuscript presented in an intelligible fashion and written in standard English?

Reviewer #1: Yes

Reviewer #2: No

5. Review Comments to the Author

Reviewer #1: This single-center retrospective cohort study over a 10-year period (2010-2019) conducted in the Cape Coast Teaching Hospital showed the mortality/survival rates in the representative NICU in Ghana and some associating factors. This is an important paper, because that might provide an important local epidemiological information for clinical decision making even in resource-constrained countries.

However, to accept the manuscript, there are still concerns about description that the authors should correct or give some comments on.

Major problems

1. Introduction part is too long to be accepted. Authors should focus on what is the background and what the aim of their study is. They should describe them more clearly.

2. Results of Demographic and clinical characteristics

1) Among 2,254 preterm babies, the authors pick up just 1,282 infants for the calculation of the prevalence of live preterm babies, because there was no precise information in 623 from referring facilities and 349 who don’t have any data of birth place. Then they might have excluded so many infants for calculation. I am not sure whether the prevalence of live preterm babies is meaningful data because there might be live preterm babies even among 623 and 349 excluded babies.

2) The number of live preterm babies were 1282, while that of survived were 1523.

I am wondering why the number of survived is larger than that of live preterm babies.

3) To solve those problems, it would be better to make a chart of patient collection.

3. Table 1

1) The authors calculated % in each variable. but it is hard to understand at first looking, because it is not easy to find out how to calculate them. For example, In Male of “Sex of Baby”, they calculated 29.7% in Died, while 70.3% in Discharged alive, and finally sum of % is 100%. On the other hand, total number of Male should be 1115. But sum of 315 in Died and 746 in Discharged alive is not 1115, which makes me confused.

2) Page13: According to APGAR Score at 1 min in Table1, the prevalence of APGAR Score at 1min<4 is higher in “Discharged alive” than “Died” (136/355(38%) vs 205/355 (58%)). Is that true? The title of Table1 is “the Factors influencing the Survival of Preterm Babies”, and so your results mean that lower APGAR Score at 1min is related to higher survival rate.

4. Table3

1) In Table 3, the authors examined factors associated with the Survival of preterm babies and showed the Adjusted OR (95%CI). But I am afraid that their interpretation of results might be wrong.

2) According to Adjusted OR in Weight of Baby with Extremely low as reference (Page 16-17), increased birth weight is correlated with decreased survival rate. Moreover, according to Adjusted OR in APGAR Score at 5min with Score 1 as reference (Page 17-18), increased APGAR Score at 5 min is correlated with decreased survival rate. Those results should not make sense.

3) The results seem to be associated with death of preterm birth, not survival.

Minor problems

1. Page 9: Data collection/handling

The authors performed logistic regression to test for association between survival and its influencing factors. They should clarify with which factors they adjusted analysis.

2. Page10: P=3.06 should be mistyping.

Reviewer #2: General Comments:

This paper reveals the survival rate of preterm infants in Ghana over a 10-year period. Although the prevalence of preterm infants is gradually increasing, survival rates have not changed significantly over the past decade. The lack of mechanical ventilators, non-invasive ventilators, syringe pumps, and blood gas analyzers clearly increases neonatal mortality rates. In the future, the introduction of these devices will improve mortality rates in Ghana. The degree of improvement in neonatal mortality rates may be used to evaluate the cost effectiveness of implemented low-cost health care resources.

Specific recommendations for revision minor:

1. Regarding the following two points:

a) On page 5 “In Ghana, there is a strong association between preterm births and advanced maternal age (>35 years), suboptimal antenatal care, preeclampsia, antepartum haemorrhage, and preterm prelabour rupture of membranes (PPROM).”

b) On page 10 “An increasing trend in prevalence is observed, with 2019 recording 9%, which is the highest for the study period (271/3027). [Figure 1.0].”

The causes of the increased rate of preterm births in Ghana need to be considered. If advanced maternal age pregnancies are the cause, the older the mother is, the higher the child mortality rate should be. However, in this paper, there is no association between child mortality and maternal age.

2. There are small grammatical errors. Examples are listed below:

On page 2:

a) Special Care Baby unit: Special Care Baby Unit

b) Data on 2,254 babies that met the inclusion criteria was extracted: Data on 2,254 babies that met the inclusion criteria were extracted

c) Descriptive statistics were generated and test of association done with chi-square and multivariable logistic regression: Descriptive statistics were generated and tests of association done with chi-square and multivariable logistic regression

d) An increasing trend in prevalence is observed, with 2019 recording the highest of 9% (271/3027).: An increasing trend in prevalence was observed, with 2019 recording the highest at 9% (271/3027).

On page 3:

e) It also highlighted the problem of inadequate data capture: It also highlights the problem of inadequate data capture

On page 7:

f) This study, therefore, will be the first to report on survival of preterm babies: This study, therefore, will be the first to report on the survival of preterm babies

On page 8:

g) which lacks some of the logistics of neonatal intensive care unit (NICU): which lacks some of the logistics of a neonatal intensive care unit (NICU)

On page 9:

h) Data on maternal age, parity, and mode and year of delivery, sex and weight of baby, gestational age, and 1st and 5th minute APGAR scores, length of stay and outcome (whether discharged alive or demised) were collected: Data on maternal age, parity, mode and year of delivery, sex and weight of the baby, gestational age, 1st and 5th minute APGAR scores, length of stay, and outcome (whether discharged alive or demised) were collected

On page 10:

i) whiles data on place of birth of 349 babies were not documented.: while data on the place of birth of 349 babies were not documented.

j) There was no statistically significant relationship between survival and the sex of a neonate (p = 3.06), age of mother (p = 0.598), parity of mother (p = 0.717), year of delivery (p = 0.212) [Table1].: There was no statistically significant relationship between survival and the sex of a neonate (p = 3.06), age of mother (p = 0.598), parity of mother (p = 0.717), or year of delivery (p = 0.212) [Table1].

On page 11:

k) However, the weight of a baby (p = <0.001), referring point (p <0.001),: However, the weight of the baby (p = <0.001), referral point (p <0.001),

On page 21:

l) as well as facilitate antenatal counselling on high-risk infants.: as well as facilitate antenatal counselling in high-risk infants.

On page 22:

m) whiles an overall singleton preterm survival rate of 38.4% has been reported in Nigeria. [26]: while an overall singleton preterm survival rate of 38.4% has been reported in Nigeria. [26]

n) such as provision of warmth,: such as the provision of warmth,

On page 23:

o) Survival of preterm babies is comparatively very high in resource rich countries due to the readily-available technological equipment.: The survival of preterm babies is comparatively very high in resource rich countries due to readily-available technological equipment.

p) Mortality rates is 52%: Mortality rates are 52%

On page 24:

q) consumer-satisfaction: patient-satisfaction

r) We found an increasing prevalence of live preterm birth at CCTH,: We found an increasing prevalence of live preterm births at CCTH,

s) Regarding the characteristics of the population, there was no significant difference between sex of baby and mortality.: Regarding the characteristics of the population, there was no significant difference between the sex of the baby and mortality.

t) Thus, we may have erroneously and inadvertently excluded some preterm babies in this study.: Thus, we may have erroneously and inadvertently excluded some preterm babies from this study.

u) Information gaps in developing countries has been highlighted in a recent review on preterm births, as one of the challenges in assessing accurate epidemiological data.: Information gaps in developing countries have been highlighted in a recent review on preterm births, as one of the challenges in assessing accurate epidemiological data.

On page 25:

v) Our data shows that preterm infants who survive the first six postnatal days have considerable chance of survival until discharge.: Our data shows that preterm infants who survive the first six postnatal days have a considerable chance of survival until discharge.

6. PLOS authors have the option to publish the peer review history of their article (what does this mean?). If published, this will include your full peer review and any attached files.

Reviewer #1: No

Reviewer #2: **Yes: **Mitsuhiro Okamoto

---

## [Author Response · Author response to Decision Letter 0]

20 Nov 2020

RESPONSE TO REVIEWER #1 COMMENTS

1. Introduction part is too long to be accepted. Authors should focus on what is the background and what the aim of their study is. They should describe them more clearly.

Response: The ‘introduction’ has been significantly revised and shortened. The study aim is also more clearly outlined.

2. Among 2,254 preterm babies, the authors pick up just 1,282 infants for the calculation of the prevalence of live preterm babies, because there was no precise information in 623 from referring facilities and 349 who don’t have any data of birth place. Then they might have excluded so many infants for calculation. I am not sure whether the prevalence of live preterm babies is meaningful data because there might be live preterm babies even among 623 and 349 excluded babies.

Response: The prevalence of live preterm births presented was for deliveries at the Cape Coast Teaching Hospital ALONE. The 623 were from other lower-level referring facilities and do not make any contribution to the prevalence measure.

It is possible there may have been some live preterm babies among the 349 for whom data on place of delivery was not documented as some of them may have been delivered at CCTH. It is again possible this could shoot up the prevalence as suggested but we cannot truly define what its impact would be. It would be acknowledged as a limitation and discussed.

Under ‘Results’, ‘Demographic and clinical characteristics of participants’ ……….the first paragraph originally read as;

“The study included a total of 2,254 preterm babies seen at the SCBU from 2010 to 2019. The CCTH had a total of 27,320 deliveries within that same period, out of which 1,282 were live preterm births, and 623 were from referring facilities whose total deliveries are not known, whiles data on place of birth of 349 babies were not documented. Thus, the prevalence of live preterm babies over the 10-year period at the CCTH alone was 4.7% (1,282/27,320). An increasing trend in prevalence is observed, with 2019 recording 9%, which is the highest for the study period (271/3027). [Figure 1.0].”

It is now revised to read as;

“The study included a total of 2,254 preterm babies seen at the SCBU from 2010 to 2019. The CCTH had a total of 27,320 deliveries within that same period, out of which 1,282 were live preterm births. A further 623 preterms were from different referring facilities while data on the place of birth for another 349 babies were not documented. Thus, the prevalence of live preterm babies over the 10-year period for deliveries at CCTH alone, was 4.7% (1,282/27,320)”

3. The number of live preterm babies were 1282, while that of survived were 1523.

I am wondering why the number of survived is larger than that of live preterm babies.

Response: There is no confusion at all with both figures. The 1282 are the number of preterm babies delivered alive at the Cape Coast Teaching Hospital over the 10 year review period. The 1523, on the other hand, refer to the number of preterm babies discharged alive from the SCBU of the Cape Coast Teaching Hospital. The latter figure thus combines some of those delivered at CCTH and some of those referred from elsewhere to CCTH.

4. Table 1

1) The authors calculated % in each variable. but it is hard to understand at first looking, because it is not easy to find out how to calculate them. For example, In Male of “Sex of Baby”, they calculated 29.7% in Died, while 70.3% in Discharged alive, and finally sum of % is 100%. On the other hand, total number of Male should be 1115. But sum of 315 in Died and 746 in Discharged alive is not 1115, which makes me confused.

Response: We acknowledge the lack of clarity in Table 1. It has therefore be redone and presented more succinctly. 

2) Page13: According to APGAR Score at 1 min in Table1, the prevalence of APGAR Score at 1min<4 is higher in “Discharged alive” than “Died” (136/355(38%) vs 205/355 (58%)). Is that true? The title of Table1 is “the Factors influencing the Survival of Preterm Babies”, and so your results mean that lower Apgar score at 1min is related to higher survival rate.

Response: We have presented results as obtained from our analysis.The results of the Chi square analysis for association between the categorical independent variable ‘Apgar Score at 1 min’ and categorical dependent variable ‘survival outcomes’ is now presented in Table 2. While the numbers pertaining to scores less than 4 are higher in the ‘discharged alive’ sub-category of the dependent variable, much higher numbers are found for scores higher than 4 and these underlie the significant association between ‘Apgar Score at 1 min’ and ‘survival outcomes’

5.Table3 

1) In Table 3, the authors examined factors associated with the Survival of preterm babies and showed the Adjusted OR (95%CI). But I am afraid that their interpretation of results might be wrong.

2) According to Adjusted OR in Weight of Baby with Extremely low as reference (Page 16-17), increased birth weight is correlated with decreased survival rate. Moreover, according to Adjusted OR in APGAR Score at 5min with Score 1 as reference (Page 17-18), increased APGAR Score at 5 min is correlated with decreased survival rate. Those results should not make sense.

3) The results seem to be associated with death of preterm birth, not survival.

Response: Table 3 has been redone and the correlates speak to survival. For instance, increasing birth weight is correlated with higher odds of survival and so is the Apgar Score at 5 min.

RESPONSE TO REVIEWER #2 COMMENTS

1. Regarding the following two points:

a) On page 5 “In Ghana, there is a strong association between preterm births and advanced maternal age (>35 years), suboptimal antenatal care, preeclampsia, antepartum haemorrhage, and preterm prelabour rupture of membranes (PPROM).”

b) On page 10 “An increasing trend in prevalence is observed, with 2019 recording 9%, which is the highest for the study period (271/3027). [Figure 1.0].”

The causes of the increased rate of preterm births in Ghana need to be considered. If advanced maternal age pregnancies are the cause, the older the mother is, the higher the child mortality rate should be. However, in this paper, there is no association between child mortality and maternal age.

Response: The statements referred to are from previous studies in Ghana and relate the incidence of preterm births to the listed factors including advanced maternal age. In the present study, we have sought to examine correlates between maternal age and survival of preterm infants ( a different phenomenon from incidence of preterm births). Our study did not find an association between maternal age and the preterm deaths/survival. There could be a number of reasons for this such as fewer numbers of women older than 35 years. It could also be that some confounders are at play here. The lack of an association will be discussed in the context of existing literature review.

2. There are small grammatical errors. Examples are listed below:

On page 2:

a) Special Care Baby unit: Special Care Baby Unit

b) Data on 2,254 babies that met the inclusion criteria was extracted: Data on 2,254 babies that met the inclusion criteria were extracted

c) Descriptive statistics were generated and test of association done with chi-square and multivariable logistic regression: Descriptive statistics were generated and tests of association done with chi-square and multivariable logistic regression

d) An increasing trend in prevalence is observed, with 2019 recording the highest of 9% (271/3027).: An increasing trend in prevalence was observed, with 2019 recording the highest at 9% (271/3027).

Response: These have been rectified as suggested.

3.On page 3:

e) It also highlighted the problem of inadequate data capture: It also highlights the problem of inadequate data capture

Response: Correction done as suggested

4.On page 7:

f) This study, therefore, will be the first to report on survival of preterm babies: This study, therefore, will be the first to report on the survival of preterm babies

Response: This has been revised to read as;

“This study is the first to report on survival of preterm babies at a specialised neonatal care unit in Ghana.”

5.On page 8:

g) which lacks some of the logistics of neonatal intensive care unit (NICU): which lacks some of the logistics of a neonatal intensive care unit (NICU)

Response: correction done as suggested

6.On page 9:

h) Data on maternal age, parity, and mode and year of delivery, sex and weight of baby, gestational age, and 1st and 5th minute APGAR scores, length of stay and outcome (whether discharged alive or demised) were collected: Data on maternal age, parity, mode and year of delivery, sex and weight of the baby, gestational age, 1st and 5th minute APGAR scores, length of stay, and outcome (whether discharged alive or demised) were collected

Response: corrections done as suggested

7.On page 10:

i) whiles data on place of birth of 349 babies were not documented.: while data on the place of birth of 349 babies were not documented.

Response: corrected and reworded as;

“…………………………. while data on the place of birth for another 349 babies were not documented.”

8.There was no statistically significant relationship between survival and the sex of a neonate (p = 3.06), age of mother (p = 0.598), parity of mother (p = 0.717), year of delivery (p = 0.212) [Table1].: There was no statistically significant relationship between survival and the sex of a neonate (p = 3.06), age of mother (p = 0.598), parity of mother (p = 0.717), or year of delivery (p = 0.212) [Table1].

Response: The above has been revised to read as;

“survival of the preterm neonate was independent of the sex of baby, maternal age and parity (see Table 1).”

9.On page 11:

k) However, the weight of a baby (p = <0.001), referring point (p <0.001),: However, the weight of the baby (p = <0.001), referral point (p <0.001),

Response: correction done as suggested

10.On page 21:

l) as well as facilitate antenatal counselling on high-risk infants.: as well as facilitate antenatal counselling in high-risk infants.

Response: correction done as suggested

11.On page 22:

m) whiles an overall singleton preterm survival rate of 38.4% has been reported in Nigeria. [26]: while an overall singleton preterm survival rate of 38.4% has been reported in Nigeria. [26]

Response: corrected as suggested

12.On page 23:

o) Survival of preterm babies is comparatively very high in resource rich countries due to the readily-available technological equipment.: The survival of preterm babies is comparatively very high in resource rich countries due to readily-available technological equipment

Response: correction done as suggested

13.Mortality rates is 52%: Mortality rates are 52%

On page 24:

q) consumer-satisfaction: patient-satisfaction

r) We found an increasing prevalence of live preterm birth at CCTH,: We found an increasing prevalence of live preterm births at CCTH,

Response: all corrections done as suggested

14.s) Regarding the characteristics of the population, there was no significant difference between sex of baby and mortality.: Regarding the characteristics of the population, there was no significant difference between the sex of the baby and mortality.

Response: reworded to read “Regarding the characteristics of the population, there was no significant association between the sex of the baby and mortality

---

## [Decision Letter · Decision Letter 1]

5 Jan 2021

PONE-D-20-26520R1

Determinants of preterm survival in a tertiary hospital in Ghana: a ten-year review

PLOS ONE

Dear Dr. Evans Kofi Agbeno

Thank you for submitting your manuscript to PLOS ONE. After careful consideration, we feel that it has merit but does not fully meet PLOS ONE’s publication criteria as it currently stands. Therefore, we invite you to submit a revised version of the manuscript that addresses the points raised during the review process.

ACADEMIC EDITOR:

Please correct the minor point suggested by reviewer, the corrected version could be accepted after the check by editorial office.

We look forward to receiving your revised manuscript.

Kind regards,

Kazumichi Fujioka

Academic Editor

PLOS ONE

Reviewers' comments:

Reviewer's Responses to Questions

**Comments to the Author**

1. If the authors have adequately addressed your comments raised in a previous round of review and you feel that this manuscript is now acceptable for publication, you may indicate that here to bypass the “Comments to the Author” section, enter your conflict of interest statement in the “Confidential to Editor” section, and submit your "Accept" recommendation.

Reviewer #1: (No Response)

Reviewer #2: All comments have been addressed

2. Is the manuscript technically sound, and do the data support the conclusions?

Reviewer #1: Yes

Reviewer #2: Yes

3. Has the statistical analysis been performed appropriately and rigorously? 

Reviewer #1: Yes

Reviewer #2: Yes

4. Have the authors made all data underlying the findings in their manuscript fully available?

Reviewer #1: Yes

Reviewer #2: Yes

5. Is the manuscript presented in an intelligible fashion and written in standard English?

Reviewer #1: Yes

Reviewer #2: Yes

6. Review Comments to the Author

Reviewer #1: Thank you for your revised manuscript entitled “Determinants of preterm survival in a tertiary hospital in Ghana: a ten-year review”

The revised manuscript shows much improvement by modifying the Tables, and I accept it after they revise two minor problems as below.

Minor problems

1. pp10, % of caesarean section;

In the revised manuscript, you showed the % of caesarean section among the 2254 babies as 45.5%. That is not correct. That should be 44.5%.

2. Table1, Survival pf preterm neonates:

% of death should be 27.6%, not 29.0%. % of discharged/Alive should be 67.6%, not 71.0%.

Reviewer #2: (No Response)

7. PLOS authors have the option to publish the peer review history of their article (what does this mean?). If published, this will include your full peer review and any attached files.

Reviewer #1: No

Reviewer #2: **Yes: **Mitsuhiro Okamoto

---

## [Author Response · Author response to Decision Letter 1]

9 Jan 2021

Reviewer #1: Thank you for your revised manuscript entitled “Determinants of preterm survival in a tertiary hospital in Ghana: a ten-year review”

The revised manuscript shows much improvement by modifying the Tables, and I accept it after they revise two minor problems as below.

Minor problems

1. pp10, % of caesarean section;

In the revised manuscript, you showed the % of caesarean section among the 2254 babies as 45.5%. That is not correct. That should be 44.5%.

Response: We appreciate your keen sense of detail. This was a typographical error on our part.

Under Results/ 2nd paragraph, the 1st sentence of page 10 originally read as;

“Of those with the mode of delivery documented among the 2254 babies, 55.5% (1,100/1983) were vaginal and 45.5% (883/1983) were by caesarean section (see Table 1).”

It has now been corrected to read as below;

“Of those with the mode of delivery documented among the 2254 babies, 55.5% (1,100/1983) were vaginal and 44.5% (883/1983) were by caesarean section (see Table 1).”

2. Table1, Survival pf preterm neonates:

% of death should be 27.6%, not 29.0%. % of discharged/Alive should be 67.6%, not 71.0%.

Response: For this variable, data was available for 2144 preterm neonates AND NOT the total of 2254. The denominator is thus 2144 (this is shown beside the variable in the table). The percentages reported therefore stand correct (approximated to one decimal place).

Editor's comment: Please amend your current ethics statement to include the full name of the ethics committee/institutional review board(s) that approved your specific study

Response: The 'ethical review board of CCTH' is now amended to read 'Ethical Review Board of the Cape Coast Teaching Hospital'

---

## [Editor Report · Decision Letter 2]

12 Jan 2021

Determinants of preterm survival in a tertiary hospital in Ghana: a ten-year review

PONE-D-20-26520R2

Dear Dr. Evans Kofi Agbeno,

We’re pleased to inform you that your manuscript has been judged scientifically suitable for publication and will be formally accepted for publication once it meets all outstanding technical requirements.

Kind regards,

Kazumichi Fujioka

Academic Editor

PLOS ONE

Additional Editor Comments (optional):

Well responded.

---

## [Editor Report · Acceptance letter]

14 Jan 2021

PONE-D-20-26520R2 

Determinants of preterm survival in a tertiary hospital in Ghana: a ten-year review 

Dear Dr. Agbeno:

I'm pleased to inform you that your manuscript has been deemed suitable for publication in PLOS ONE. Congratulations! Your manuscript is now with our production department. 

Kind regards, 

on behalf of

Dr. Kazumichi Fujioka 

Academic Editor

PLOS ONE